# Molecular Mechanisms of Maternal Diabetes Effects on Fetal and Neonatal Surfactant

**DOI:** 10.3390/children8040281

**Published:** 2021-04-06

**Authors:** Hilal Yildiz Atar, John E. Baatz, Rita M. Ryan

**Affiliations:** 1Departments of Pediatrics (Neonatology), UH Rainbow Babies and Children’s Hospital, Case Western Reserve University, Cleveland, OH 44106, USA; Rita.ryan@uhhospitals.org; 2Departments of Pediatrics (Neonatology), Medical University of South Carolina, Charleston, SC 29425, USA; baatzje@musc.edu

**Keywords:** surfactant protein, surfactant lipids, infants of a diabetic mother, respiratory distress syndrome, lung development, hyperglycemia

## Abstract

Respiratory distress is a significant contributor to newborn morbidity and mortality. An association between infants of diabetic mothers (IDMs) and respiratory distress syndrome (RDS) has been well recognized for decades. As obesity and diabetes prevalence have increased over the past several decades, more women are overweight and diabetic in the first trimester, and many more pregnant women are diagnosed with gestational diabetes. Glycemic control during pregnancy can be challenging due to the maternal need for higher caloric intake and higher insulin resistance. Surfactant is a complex molecule at the alveolar air–liquid interface that reduces surface tension. Impaired surfactant synthesis is the primary etiology of RDS. In vitro cell line studies, in vivo animal studies with diabetic rat offspring, and clinical studies suggest hyperglycemia and hyperinsulinemia can disrupt surfactant lipid and protein synthesis, causing delayed maturation in surfactant in IDMs. A better understanding of the molecular mechanisms responsible for surfactant dysfunction in IDMs may improve clinical strategies to prevent diabetes-related complications and improve neonatal outcomes.

## 1. Diabetes and Maternal/Fetal Health

The prevalence of type 1 diabetes mellitus (DM), type 2 DM, and gestational DM (GDM) have increased dramatically in the last 20 years [1,2]. As its prevalence continues to grow, diabetes is becoming a significant health problem and one of the most common diseases that adversely affects maternal and fetal health. According to the most recent International Diabetes Federation data in 2019 [3], one in six live births is affected by hyperglycemia during pregnancy. Among those pregnancies, 84% of women have GDM. Thus, annually 20 million babies are exposed to hyperglycemia in utero.

Glucose homeostasis affects fetal growth and development throughout pregnancy [4]. Insulin resistance (IR) in the mother increases physiologically during pregnancy, especially in the last trimester [5]. Increased IR results in maternal energy coming more from fat metabolism, sparing carbohydrate usage by the rapidly growing fetus [5]. During pregnancy, this physiologic state of IR increases the risk in women who already have insulin resistance to have more imbalanced glucose homeostasis, leading to GDM or worsening pregestational diabetes (PGD) [6]. Studies have shown that women diagnosed with GDM already have some degree of IR [7]. Up to 60% of those women develop type 2 DM later in life [8].

O’Sullivan described the necessity of glucose tolerance testing during pregnancy in the 1960s, and also developed the first diagnostic criteria for GDM in 1964 [9]. The diagnostic criteria have been revised multiple times since then. Most recently, the American Diabetes Association recommends using either the one- or two-step approach at 24–28 weeks of gestation to test mothers, who do not previously have diabetes to assess for the presence of GDM [10]. Clinical studies using maternal glycosylated hemoglobin (HbA_1C_) as a diagnostic tool for GDM are controversial [11,12,13]. HbA_1C_ is significantly lower in pregnancy than in nonpregnant women, and fasting glucose is lower in early pregnancy [14]. Establishing diagnostic criteria for HbA_1C_ during pregnancy might reduce the need for an oral glucose tolerance test (OGTT) among pregnant women and perhaps be easier to test as it can be a one-step test, requiring less time than oral OGTT. Randomized controlled trials in larger populations could improve our understanding of the role of HbA_1C_ during pregnancy. As we do not have a clear answer on how to use HbA_1C_ during pregnancy, OGTT remains the preferred diagnostic test for GDM.

There is a strong association between impaired glucose tolerance and diabetes during pregnancy with multiple fetal congenital anomalies, indicating that maternal hyperglycemia may be a significant teratogen to the growing fetus [15,16]. Maternal HbA_1C_ correlation with congenital malformations was found in an analysis of seven cohort studies from 1997 pregnancies [17]. These pregnancies resulted in 117 live births with congenital anomalies, and maternal HbA_1C_ ≥14% resulted in a 20% congenital malformation rate, while an HbA_1C_ of 7.6% had a congenital malformation rate of approximately 4%. The other most common problems in infants of diabetic mothers (IDMs) include, but are not limited to, hypoglycemia, hypocalcemia, hyperbilirubinemia, hyperinsulinism, macrosomia, respiratory distress syndrome (RDS), preterm delivery, cardiac anomalies, including diabetic cardiomyopathy, caudal regression syndrome, and small colon syndrome [18,19].

## 2. Infants of a Diabetic Mother (IDMs) and RDS

Respiratory distress syndrome (RDS) is a common cause of respiratory distress affecting newborns. RDS occurs secondary to surfactant deficiency due to inadequate production of surfactant. Avery and Mead first discovered the link between surfactant deficiency and clinical RDS (called hyaline membrane disease at that time) in the 1950s [20]. In 1959 it was first described by Gellis and Hsia that IDMs had increased mortality and morbidity due to RDS [21]. Since that time, the effects of maternal diabetes on the fetus have become an extensive research area of interest. A retrospective analysis by Robert, et al. showed that after controlling for other confounder factors, including gestational age (GA) and delivery route, IDMs have a 5.6 times greater risk of developing RDS than those infants of nondiabetic gestation [22].

In a recent prospective study of late preterm infants born to a mother with GDM [23], the authors noted that GDM mothers were statistically older and had higher body mass index at the time of the delivery. Severe RDS in this study was defined as clinical signs of early respiratory distress occurring within the first two hours following birth, with consistent radiologic features and oxygen dependence requiring invasive and/or noninvasive mechanical ventilation with a fraction of inspired oxygen (FIO_2_) >0.25 for a minimum of 24 h and admission to a neonatal intensive care unit (NICU). GDM was found to be a significant risk factor for severe RDS. UK’s Confidential Enquiry into Maternal and Child Health (CEMACH) study between 2002 and 2007 was the largest study to date to investigate outcomes of pregnant women with type 1 and type 2 DM [24]. In this study, neonatal outcomes of macrosomia, RDS, and shoulder dystocia were not significantly different between maternal type 1 vs. type 2 DM. The most crucial factor for adverse neonatal effects was thought to be high maternal glucose concentration [24]. A systematic meta-analysis of studies performed throughout 1987–2008 compared fetal outcomes between type 1 and type 2 DM (total of 3781 and 7966 pregnancies, respectively) and did not reveal any statistically significant difference in RDS [25]. These results suggest that the type of diabetes does not influence RDS outcome.

A small prospective study with 18 type 1 DM pregnant women was designed to show improvement in maternal euglycemia with continuous subcutaneous glucose monitoring and continuous insulin administration [26]. Glucose monitoring was performed at two different occasions where diabetic women are prone to be hyperglycemic: 72 h after betamethasone administration and during labor. Infants were observed for hypoglycemia and RDS as primary outcomes; none of them had hypoglycemia or RDS. Even though preterm infants <34 weeks GA are more at risk of having RDS, preterm infants in this study did not show RDS, suggesting better glycemic control in diabetic pregnancies can improve the neonatal outcome. Unfortunately, this was a small study without a proper control group, and this question would benefit from a larger randomized controlled trial (RCT).

## 3. Delayed Lung Maturation in IDMs

### 3.1. Effects on Surfactant Phospholipid Composition

Pulmonary surfactant is a complex molecule with a mixture of lipids (90%) and protein (10%), produced by type II alveolar epithelial cells (AEC2s) [27]. The function of pulmonary surfactant is to decrease the alveolar surface tension to increase lung compliance and prevent alveolar collapse at the end of expiration [27]. The primary surfactant lipid components are phosphatidylcholine (PC), phosphatidylglycerol (PG), and phosphatidylinositol (PI). PC constitutes approximately 70% of the lipid portion of surfactant, and it exists primarily in an unsaturated form known as dipalmitoyl phosphatidylcholine (DPPC) [28]. Surfactant is packaged and stored in large intracellular inclusions named lamellar bodies. Although other cells can secrete some surfactant components, the AEC2 is the only cell that regulates the secretion and storage of functional surfactant [29].

AEC2s start producing surfactant at ~24 weeks of GA. Historically, it is known that there is a net efflux of lung fluid into the amniotic fluid (AF). Hence, AF has been utilized to understand lung maturation. Fetal lung maturation (FLM) testing was first discovered by Gluck in 1971 [30], used to predict the risk of RDS, and has been used clinically and in research for years. FLM testing estimates fetal lung maturity by measuring the presence and/or concentration of the surfactant components in the AF obtained by amniocentesis. Due to the widespread use of early first-trimester ultrasound for pregnancy dating, FLM testing is less needed in current practice to predict the risk of having RDS following birth [31]; however, it provides insight about in utero FLM. Significant components of FLM are the lecithin (PC)/sphingomyelin (L/S) ratio, presence of phosphatidylglycerol (PG), disaturated phosphatidylcholine (DSPC), and lamellar body count. It is well known that the phospholipid composition changes over gestation in all air-breathing species [32]. As term gestation approaches, PG increases, phosphatidylinositol (PI) decreases [33], and the L/S ratio increases [34].

Infants with RDS have an absent or very low level of PG [35,36,37]. Diabetic pregnancies were associated with delayed PG production compared with the nondiabetic control group [38]. A case-control study comparing AF surfactant phospholipids between control groups and diabetic pregnancies examined 981 amniocenteses performed for FLM [39]. Of these pregnancies, 372 of the women had diabetes (74% GDM; 26% PGD). Their AF did not show any difference in L/S ratio but did show delayed PG production by 7–10 days in diabetic pregnancies regardless of the type of diabetes. Although GDM patients showed more significant delay than PGD patients, the difference was not statistically significant. The PI level difference was substantial: PI level in AF from PGD pregnancies peaked to higher levels in an earlier GA than in nondiabetic control groups; however, GDM pregnancies failed to show this effect. As in other studies, patients with diabetes had a significantly higher C-section delivery rate; their babies had a higher NICU admission rate and a significantly higher birth weight of approximately 400 g. Hallman et al. noted similar data with diabetic vs. nondiabetic pregnancies: in AF from diabetic patients, PG was absent or significantly lower, and PI remained high with no difference in L/S between the two groups [40]. Thus, maternal diabetes is associated with a delay in the changeover from PI to PG in surfactant phospholipid composition (Figure 1). The effect of glycemic control (good vs. poor; good glycemic control defined as HbA_1C_ < 5.8) on FLM was prospectively studied in 621 pregnant diabetic women (511 GDM; 110 PGD women) [41]. A significant difference in levels of PG in near-term AF samples was observed in poor-glycemic-control pregnancies compared to good-glycemic-control pregnancies. Consistent with these findings and those of previous studies, there was no significant difference in surfactant components based on the types of maternal diabetes. The exact mechanism for the evolution from less PI in surfactant and higher PG is not known.

Glucose is an essential substrate for surfactant lipid synthesis. Insulin not only regulates glucose uptake to cells but also regulates surfactant synthesis [49]. The effect of different insulin and glucose concentrations on glucose uptake, glucose metabolism, and surfactant synthesis were examined in AEC2 cultures by Engle et al. [50]. AEC2s derived from fetal rat lung at 19 days of gestation (term = 21) were cultured in different insulin and glucose concentrations. The addition of 10 units/mL of insulin caused a 35% increase in surfactant PC synthesis. However, 100 units/mL insulin reduced PC synthesis to below control levels. The exposure to insulin (3 h vs. 24 h) did not change the result. These results indicate that a physiological level of insulin plays a role as a stimulatory hormone in surfactant synthesis, but a high insulin level can inhibit surfactant PC synthesis (Figure 2). The effect of hyperglycemia in surfactant lipid production was further studied in fetal rat lung excipients to investigate if there is a critical time period in pregnancy during which AEC2s are more sensitive to hyperglycemia [51]. Fetal rat lung explants from embryonic day 18–22 were exposed to high glucose (100 mM). Increased choline incorporation into PC and DSPC was observed on days 18–19 and in the high glucose exposure group and significantly decreased on days 20–22. The total amount of PC and DSPC significantly decreased on day 20 and did not show any difference on day 18–19. These results suggest that AEC2 from late preterm neonatal rat lung explants was more sensitive to hyperglycemia and not only had impaired glucose utilization but also had less PC and DSPC.

In addition to the glucose being an important substrate for surfactant lipid synthesis, glycogen storage in AEC2 is also crucial [55]. AEC2 glycogen increases with increased gestational age, followed by a rapid decline later in pregnancy, and correlates with the acceleration of PC and surfactant synthesis [56]. A rapid decrease in AEC2 glycogen after maternal betamethasone administration in animal studies suggests that glycogen degradation is pivotal for surfactant synthesis and positively correlates with lung maturation [57,58]. Glycogen phosphorylase, the enzyme that degrades glycogen, exists in two forms: “A” is the active (dephosphorylated form) and “B” is the inactive (phosphorylated form). An increase in glycogen phosphorylase A activity is related to increased degradation of glycogen storage. Pulmonary glycogen storage and degradation were studied in streptozotocin-induced diabetic rats (STZ-DB), a well-established model for diabetic pregnancy and their offspring [47]. STZ-DB fetuses failed to decline in their glycogen storage compared to the control group in late gestation on days 21–22 (term = 22). Glycogen phosphorylase A activity also was unable to increase and was significantly lower compared with nondiabetic control groups. These findings were consistent with glycogen underutilization, causing impaired surfactant synthesis leading to delayed lung maturation in diabetic pregnancies. Glucose inhibits phosphorylase A activity by increasing phosphorylate phosphatase, which decreases glycogen phosphorylase A activity [59]; hyperglycemia causing a marked reduction in glycogen phosphorylase A activity (Figure 2); however, the effect of hyperinsulinism on phosphorylase A activity is not apparent [47]. The animal model used in this study does not fully mimic IDM due to the lack of hyperinsulinism. It is unclear how this pathway works in humans, and additional research would improve our understanding of the association between glycogen degradation and RDS in IDMs.

In summary, maternal DM is a significant risk factor for altered surfactant lipid production in fetuses, and the type of DM does not seem to have a substantial effect on it.

### 3.2. Effects on Surfactant Protein Composition

Although phospholipid is the major component of surfactant, the four surfactant-associated proteins play critical roles in surfactant synthesis and function [60]. These four proteins are designated as surfactant protein A (SP-A), SP-B, SP-C, SP-D. These proteins can be divided into two groups based on their structure and amino acid compositions: SP-B and SP-C are two very small (8 and 4 kDa) [28,61] hydrophobic proteins that are essential for normal surface tension-lowering ability of surfactant and packaging of lamellar bodies. SP-A and SP-D are larger (35 and 43 kDa) hydrophilic proteins [62,63] and part of the innate immune system of the lung in addition to their roles in surfactant hemostasis [64].

The critical role of SP-B was solidified by recognition of a rare congenital SP-B deficiency in full-term infants who died shortly after birth due to severe respiratory failure resembling severe RDS [65]. SP-B knockout mice were created in 1996, and not only do they lack SP-B, but their SP-C is abnormal [66]. As anticipated, the SP-B knockout mice die shortly after birth due to respiratory failure [67]. SP-C knockout mice have a normal SP-B level, and they do not have significant respiratory distress, confirming the need for only one of the two hydrophobic proteins for normal surfactant function [68]. SP-C knockout mice have unstable surfactant at lower lung volumes, suggesting an essential role of SP-C at low lung volume in RDS [69]. SP-C-deficient animal models develop fibrosis later, and autosomal dominant SP-C mutations are also associated with fibrosis in humans [70].

McGillick et al. studied the effects of hyperglycemia in late-gestation fetal sheep [42]. Control group lamb fetuses received infused saline between 130 and 140 days of gestation, and experimental group fetuses were infused glucose. Samples were obtained from animals on day 140 ± 1 (term 150 ± 1). Lung mRNA expression of glucose transporters *SLC2A1* and *SLC2A4,* and Insulin-Like Growth Factor-1 (IGF-1), and IGF-2 were not different between the groups. However, decreased IGF-1 receptor (IGF-1R) mRNA expression was seen in the lung of the glucose-infused fetus. Although the total number of SP-B-positive cells in the alveolar epithelium of the fetal lung was not different between saline- and glucose-infused fetuses, whole lung mRNA for all four surfactant proteins, measured by quantitative polymerase chain reaction (qPCR), was significantly reduced. This study provided evidence for the direct effect of high glucose on depressing SP mRNA. A reduction in all four SPs is likely to impair surfactant ability to lower alveolar surface tension and the smooth transition to extrauterine life, increasing the risk of having RDS in IDMs (Figure 1).

The effect of fetal hyperglycemia on SP-A, SP-A mRNA, SP-B, SP-B mRNA, SP-C, and SP-C mRNA levels was observed in STZ-DB offspring [43,44]: A significant reduction was observed in late gestation fetal days 18–21. Those offspring quickly recovered on neonatal days 1–2 with close to the expected levels of surfactant proteins and their associated mRNAs. In this animal model, fetal rats were hyperglycemic with a low to normal level of insulin, so these results are suggestive of specifically a hyperglycemia inhibitory effect on protein and mRNA, and not due to a hyperinsulinemia effect (Figure 2).

The effects of hyperglycemia on the production of SP-A, -B, and -C were further studied ex vivo; STZ-DB fetal rat lungs were obtained on embryonic day 20 and cultured in different concentrations of glucose: 10, 25, 50, and 100 mM [45]. While SP-B and SP-C mRNA production was significantly reduced with increased glucose concentration at 100 mM (<10%) compared to at 10 mM, SP-A mRNA was not significantly affected in various levels of glucose. These in vitro results suggested high glucose concentration inhibits the SP-B and SP-C synthesis, contradicting the previous in vivo SP-A study [43]. A separate study using the H441 cell line, a human pulmonary adenocarcinoma cell line shown to express SP-A mRNA and SP-B mRNA, supports the hypothesis of the concentration-dependent inhibitory effect of high insulin level on surfactant protein gene expression [46].

As in in vitro studies using lung cell lines and in vivo animal studies, the level of SP-A in AF is significantly reduced between 36 and 40 weeks of GA in human pregnancies complicated by diabetes compared to their GA-matched group [52]. These infants did not develop RDS, but since SP-A does not exhibit surface tension-lowering properties as detailed above, the lack of RDS is not an unexpected finding.

### 3.3. Effects on Receptors

Glucose is an essential substrate for surfactant phospholipid, and the insulin receptor regulates glucose uptake in cells [71]. Therefore, impaired glucose uptake, or insulin receptor maladaptation can play a significant role in surfactant lipid production. The insulin receptor has a complex signaling pathway mechanism involving tyrosine kinase (TK) activation [72]. The combination of high glucose and high insulin resulted in a significant reduction in insulin receptor TK activity and insulin receptor mRNA in fetal rat lung [48] (Figure 1). Downregulated receptors and TK activity resulted in significantly low glucose uptake from the culture, suggesting clinically significant insulin receptor and TK activity in glucose metabolism. After being exposed to hyperglycemia, in an immediate period of hypoglycemia, TK receptor activity remained diminished for up to 8 h and recovered after a 12 h exposure [73]. The IDM physiology is similar to that shown in this study: infants have sudden changes in their glucose levels from being hyperglycemic to being hypoglycemic after the umbilical cord is cut. Downregulated insulin TK activity receptors may not provide enough glucose to AEC2, resulting in surfactant deficiency. This acute disruption to surfactant synthesis and later recovery of receptor activity in low to normal glucose levels brings the question of whether or not a euglycemic state closer to birth can increase substrate availability in AEC2s to overcome the defective surfactant and explain the wide degree of RDS severity among IDMs.

IGF-1 regulates lung development via the IGF-1R. IGF-1R knockout mice exhibit hypoplastic lungs and lung development arrested in the pseudoglandular stage [74]. Although an increased level of IGF-1 was seen in the cord blood sample from IDMs, it is not known if IGF-1 expression is also high in the IDM lung [75]. Increased expression of IGF-1 and IGF-1R from lung autopsy samples of babies who died from RDS and bronchopulmonary dysplasia suggests balanced IGF-1/IGF-1R being very important for lung development and maturation [76].

In addition to the insulin receptor, there are other intracellular signaling mechanisms involved in surfactant synthesis, such as FOXA2. FOXA2, also known as hepatocyte nuclear factor (HNF) 3B, is a member of the forkhead box (Fox) protein family and one of the critical signaling molecules for surfactant synthesis regulation during fetal lung development [77]. Although FOXA2 plays a role in other organogenesis, such as in the pancreas, liver, and adipose tissue, the expression of FOXA2 is limited to AEC2 in the third trimester of the pregnancy when lung development and surfactant synthesis peaks in fetal life. Maternal diabetes effects on FOXA2, SP-B, and SP-C synthesis were studied in the STZ-DB rat models of preexisting diabetes and GDM [53,54]. Both groups of fetuses were derived from diabetic rats. In the diabetic offspring, fetal lung expressions of SP-B, SP-B mRNA, SP-C, SP-C mRNA, and FOXA2 in the nucleus (n-FOXA2) were significantly lower in both PGD and GDM offspring, while phosphorylated FOXA2 (which decreases FOXA2 transcription activity) in PGD, and nitrolyogenic FOXA2 (deactivated FOXA2) in GDM was significantly higher (Figure 2).

Akt/mammalian target of rapamycin (mTOR) pathway is another insulin signaling mechanism. mTOR is a serine/threonine kinase that regulates cell growth and differentiation [78]. mTOR controls insulin signaling in cellular level. The importance of the Akt/mTOR pathway was studied in a transgenic mouse model [79]. In utero, Akt activation in lung epithelial cell showed impaired maturation of the lung epithelium, downregulation of SP-B, and increased glycogen storage. Inhibition of Akt/mTOR pathway with rapamycin improved alveolar epithelial cell differentiation, decreased glycogen storage, and produced normal expression of SP-C. Downregulation of mTOR signaling mechanism possibly plays an important role in lung maturation and epithelial differentiation.

These findings suggest the importance of euglycemia and normal insulin level for the regulation of glucose uptake and surfactant synthesis.

## 4. Conclusions

Any variety of maternal diabetes (PGD type 1 or 2, or GDM) is a significant contributor to fetal health. Fetal hyperglycemia and hyperinsulinism secondary to maternal diabetes disrupt normal surfactant synthesis and function, which leads to surfactant inadequacy and clinical RDS in neonates. As maternal diabetes increases the risk of RDS in near-term infants, finding different methods to achieve better glycemic control is needed, as is further understanding of the molecular processes during GDM/PGD affecting the surfactant system. Identifying the critical factors in IDMs who do not show signs of RDS may open up other targets for the prevention and treatment of RDS in this population. As we learn more about the increased incidence of long-term health problems related to IDMs, such as hypertension, insulin resistance, diabetes, obesity, and neurodevelopmental impairment, we need to pay more attention to maternal health to improve neonatal outcomes and lifelong health.

## Figures and Tables

**Figure 1 children-08-00281-f001:**
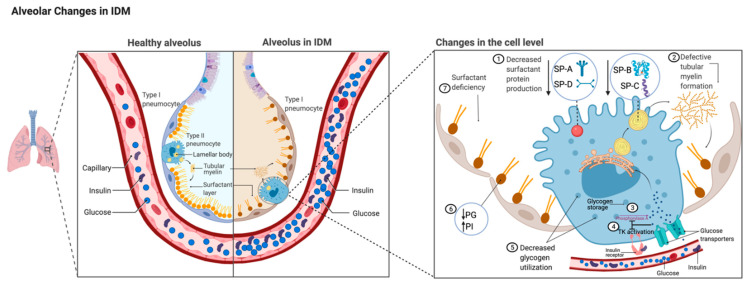
Comparison of alveolus between healthy lungs and infant of a diabetic mother (IDM) lung. Changes in IDM type II alveolar epithelial cell level magnified on the right. Fetal hyperglycemia and hyperinsulinemia secondary to maternal hyperglycemia affect surfactant production via different mechanisms. Hyperglycemia and hyperinsulinism are illustrated in the capillary. (1) Decreased level of surfactant proteins (SP-A, -B, -C, and -D) [42,43,44,45,46]. (2) Decreased level of SP-B causes abnormal tubular myelin formation. (3) Excess glucose inhibits phosphorylase A activity [47]. (4) Glucose decreases insulin receptor tyrosine kinase (TK) activity level [48]. (5) Glycogen storage is decreased due to decreased glycogen phosphorylase A activity. (6) Alterations in surfactant phospholipid synthesis affected by inadequate glucose resulting in decreased phosphatidyl glycerol (PG) and increased phosphatidyl inositol (PI) [38,39,40]. (7) These changes contribute to increased respiratory distress syndrome secondary to surfactant deficiency in IDMs.

**Figure 2 children-08-00281-f002:**
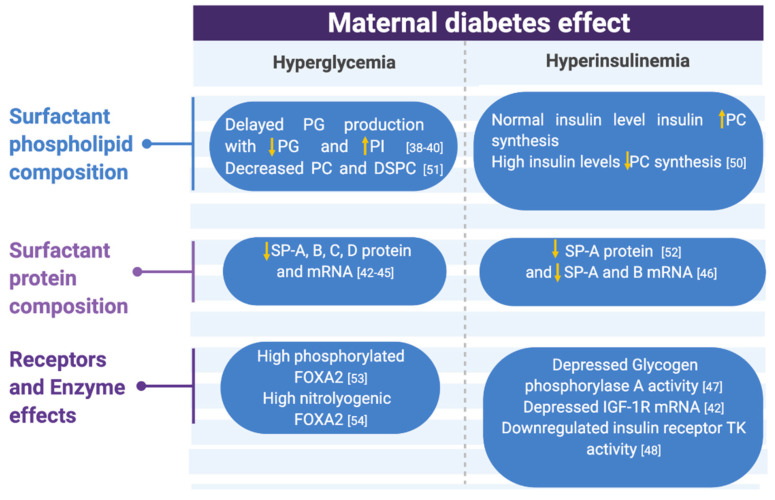
Metabolic changes associated with maternal diabetes effects on surfactant compositions, receptors, and enzymes summarized. Abbreviations: PG (phosphatidyl glycerol), PI (phosphatidyl inositol), PC (phosphatidyl choline), DSPC (desaturated phosphatidyl choline), SP (surfactant protein), IGF-1R (insulin-like growth factor 1 receptor), and TK (tyrosine kinase) [38,39,40,42,43,44,45,46,47,48,50,51,52,53,54].

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
