# Peer review of "Molecular Mechanisms of Maternal Diabetes Effects on Fetal and Neonatal Surfactant"

_children, 2021, doi:10.3390/children8040281_

Round 1

Reviewer 1 Report

In the manuscript entitled „Molecular Mechanisms of Maternal Diabetes Effects on Fetal and Neonatal Surfactant”, the Authors analyze available studies on molecular mechanisms responsible for neonatal surfactant  dysfunction in infants od diabetic mothers. The manuscript is well written and easy to read. The rationality of the review is well presented.  

The manuscript needs some minor editing changes only- e.g. the lack of space between two words - lines 268 „10units/ml”.

Author Response

Thank you for these comments.  This typographical error has been fixed.

Reviewer 2 Report

Very well written and comprehensive review of the topic. I do not have any major changes but some minor edits/suggestions.

  • Page 2, Line 45-46 "While the...GDM." The sentence is difficult to read and may need some clarification.
  • Page 2, Line 52-54. Authors may wan to consider adding need for oral GTT rather than oral GTT. 
  • Page 2. 3rd paragraph starts quite abruptly. Authors may want to consider adding a sentence on what this paragraph is about. 
  • Page 3 Line 88. Does NRDS definition has Syndrome in it? 
  • Page 4: Line 33. I think there is a typo in FML and should be FLM.
  • Page 4. Line 62. I am not sure surfactant outcomes is accurate. Consider using a different word. 

Reviewer 3 Report

Even though the artcicle is balanced and well written, I have some concerns about the literature that has been cited. Only 19/73 references that have been used in this manuscript date from the last decade. Some very recent studies on surfactant in a diabetic rat model have not been cited.

The manuscript fails to describe the distinction between problems the fets encounters with early gestational hyperglycemia (teratogenic effects and malformations) and later gestational hyperglycemia (RDS, macorsomia...). 

Figure 1 is excellent, but would improve if the text that is printed inside the figure would have a larger font size. In the current size part of the text is illegible. 

Reviewer 4 Report

Thank you for the opportunity to review this manuscript. This is a detailed review of the molecular mechanisms leading to surfactant dysfunction in IDM. Given the rapidly increasing cases of DM and its subsequent adverse effects on the neonate, this is an especially important topic for neonatologists.

The breadth and accuracy of the review is appropriate. The manuscript is generally clearly written and organized. To enhance the organization, I would suggest matching the points in figure 1 legend to how they appear within the manuscript. The studies cited are appropriate and provide a comprehensive understanding of what is currently known about IDM and molecular dysfunction of surfactant. Throughout the manuscript, the authors also highlight several areas for potential future research.

Major issues:

I did not find major issues within this review.

Minor issues:

Line 40-42, authors mention ‘multiple studies have shown…’ , however only one study is referenced.

Section 2, RDS and NRDS is used interchangeably.

Line 107-109, this sentence is slightly difficult to read. If they describe the study participants in the first sentence i.e. line 103-105 it might be better.

Line 139-140, can eliminate ‘high concentration of PG in AF has been accepted as a marker of pulmonary maturation’ as the previous paragraph is dedicated to establishing the factors that determine lung maturation.

Line 251, ‘STZ’ full form required

Line 298-300 – DM affects surfactant protein and lipid function in addition to its effect on the AEC2 receptors. Therefore, even if there was glycemic control immediately before birth it would not decrease the overall rate of RDS in IDM.

Author Response

Thank you for these comments and helpful suggestions. Please see the attachment. 

Reviewer 5 Report

Briefly, this is a complete and interesting review of the literature regarding the increased risk for RDS in infants of diabetic mothers.

Minor comments:

It would be interesting to have a bit more epidemiologic data, if available, to quantify the increased risk.  Ancestry data would be extremely informative – do the racial differences in the prevalence of RDS in premature infants persist in infants of diabetic mothers?

Since this is a review article, the authors might consider adding the figure from Avery and Mead’s classic 1959 paper that displays the surface tension measurements vs birthweight. There is one individual (triangle) that stands out with high surface tension and high birthweight – it is an infant of a diabetic mother. I think this figure is so classic that it should not be forgotten. I attached a pdf of a slide with this figure. 

The authors should review the grammar  - there are many complex sentences where tense or article do not agree or in which there are non-sequiturs.  The meaning comes through, but I had to read some sentences several times.

Just a few examples: Lines 38-40, 44-49, 52-55, 87-92    (I did not enumerate all of them…)
